# Socio-economic disparities and predictors of fertility among adolescents aged 15 to 19 in Zambia: Evidence from the Zambia Demographic and Health Survey (2018)

**Samson Shumba**[1]*, **Vanessa Moonga**[2], **Thomas Osman Miyoba**[1], **Stephen Jere**[3], **Jessy Mutale Nkonde**[1], **Peter Mumba**[4]

1 Department of Epidemiology & Biostatistics, School of Public Health, University of Zambia, Lusaka, Zambia, 2 Department of Economics, School of Business, University of Lusaka, Lusaka, Zambia, 3 Department of Monitoring, Evaluation and Learning, Avencion Limited, Lusaka, Zambia, 4 Department of Population Studies, School of Humanities and Social Sciences, University of Zambia, Lusaka, Zambia

* samsonshumba1@gmail.com

**Data Availability Statement:** Authorization to utilize the data was secured from ICF Macro,

## Abstract

Globally, 12 million girls aged 15–19 give birth each year, and Africa hosts 19% of youth aged 15–24. In Zambia, 29% of adolescents experience childbirth, with variations by age. Projections suggest a continued rise in these trends by 2030. Zambia came up with Adolescent Health Strategic Plan 2011–2015 among the specific policies being advocated for was Adolescent-Friendly Health Services (ADFHS) in order to mitigate among others adolescent fertility. The study aims to investigate socio-economic disparities and predictors of fertility in Zambian adolescents aged 15 to 19. The study used a cross-sectional study design utilized the 2018 Zambia Demographic Health Survey (ZDHS). The variable of interest in this study is "total number of children ever born" among adolescents aged 15 to 19 years. The explanatory variables that were used in the study were demographic, socio-economic, behavioral and community level factors. The Rao–Scott Chi-square test was used to test for association between categorical variables. Determinants of adolescent fertility were identified through a multilevel ordinal logistic regression conducted at a significance level of 5%. Analysis in the study was carried out using Stata version 14.2. A total of 3,000 adolescents were involved in the study, revealing that 75.88% had not given birth, 21.14% had one child, and 2.98% had at least two children. The findings revealed that education played a protective role, with adjusted odds ratios (AOR) of 0.47 (95% CI, 0.23–0.97), 0.21 (95% CI, 0.10–0.47), and 0.03 (95% CI, 0.00–0.54) for primary, secondary, and tertiary education, respectively. On the other hand, certain factors were associated with an elevated risk of fertility. These included the age of adolescents, educational attainment, marital status, wealth index, contraceptive use, exposure to family planning (FP) messages, being educated about FP at health facilities, and age at first sexual encounter. Among contextual factors, only community age at first birth was identified as a predictor of fertility, AOR, 1.59 (95% CI, 1.01–2.52). The study highlights sociodemographic disparities in adolescent fertility, emphasizing the need for targeted sexual reproductive health policies. Education protects against having more than one child, while marital status significantly influences fertility, particularly for

accessible at https://dhsprogram.com/data, under the dataset title ZMIR71DTA.

**Funding:** The authors received no specific funding for this work.

**Competing interests:** The authors have declared that no competing interests exist.

married adolescents. The research provides valuable insights into the complex factors shaping adolescent fertility in Zambia, offering guidance for interventions and policies to support this vulnerable demographic.

## Background

Every year, an estimated 21 million girls aged 15 to 19 years in developing regions become pregnant and approximately 12 million of them give birth [1]. Globally, adolescent birth rate has decreased from 64.5 births per 1000 adolescent girls (15–19 years) in 2000 to 41.3 births per 1000 adolescent girls in 2023. However, rates of change have been uneven in different regions of the world with the sharpest decline in Southern Asia and slower declines in the Latin American and Caribbean (LAC) and Sub-Saharan Africa (SSA) regions. Although declines have occurred in all regions, SSA and LAC continue to have the highest rates globally at 99.4 and 52.1 births per 1000 women, respectively in 2022. In the WHO African Region, the estimated adolescent birth rate was 97 per 1000 adolescent girls in the European Region [2].

Africa is home to the world's youngest and rapidly growing population. In 2019, the continent had an estimated 230 million young people aged between 15 and 24, constituting 19% of the global youth population. Projections indicate that by 2030, the number of youths residing in Africa will have surged by 42 percent in 2023 [3]. Alarming trends suggest that the total number of teenage pregnancies is expected to increase by 2030, with sub-Saharan Africa projected to witness a higher prevalence. Notably, the African nations with the highest prevalence of teenage pregnancies include Niger, Mali, Angola, Mozambique, Guinea, Chad, and Cote d'Ivoire [4]. This is particularly concerning given that the region already leads in both teenage pregnancies and child marriages [5].

For over four decades, Zambia has grappled with persistently high fertility rates [6, 7]. However, there has been a noteworthy decline in the total fertility rate, decreasing from 6.5 children in 1992 to 4.7 children in 2018 [7, 8]. Approximately 29% of adolescents aged 15 to 19 in Zambia have already given birth, with 6% of these births occurring among 15-year-olds and a substantial 58% among those aged 19 [6]. Significant variations exist in the percentage of adolescent girls aged 15 to 19 engaging in childbearing, ranging from 14.9% in Lusaka to 42.5% in the Southern Province in 2018 [7].

Various studies have shown that the key determinants of fertility among the older adolescents and younger adults is individual's current age, type of residential area, educational attainment, contraceptive utilization, and socioeconomic status. Research indicates that adolescent fertility often stems from the denial of sexual and reproductive health rights for young girls. Zambia grapples with one of the world's highest child marriage rates, with a slight decrease from 31.7% in 2024 to 29% in 2018 [9]. Notably, 16.5% of girls aged 15 to 19 are married, and 31.4% of those aged 20 to 24 married before turning 18 [10]. The 2007 Zambia Demographic Health Survey revealed that among the general population aged 15–19 years, 12.3% and 16.2% of women and men respectively had sexual debut before the age 15, with sexual activities comes with pregnancies and eventually children [11]. In response to critical issues like communicable and non-communicable diseases, early marriages, and pregnancies, Zambia implemented the Adolescents Health Strategic Plan (2011–2015), endorsed by the Adolescent Friendly Health Services (ADFHS), to tackle challenges, including adolescent fertility [12].

Adolescent mothers aged between 10 and 19 face higher risks of eclampsia, puerperal endometritis and systemic infections than women aged 20 to 24 years, and babies of adolescent

mothers face higher risks of low birth weight, preterm birth and severe neonatal condition. Moreover, research has underscored the correlation between age and maternal mortality, with a prevalence rate of 13% observed among individuals aged between 10 and 19 years [13]. Preventing pregnancy among adolescents and pregnancy related mortality and morbidity are foundational to achieving positive health outcomes across the life course and imperative for achieving the sustainable development goals (SDGs) related to maternal and newborn health [14]. In light of these concerning trends, this study aims to investigate the socio-economic disparities and associated factors of fertility among adolescents aged 15 to 19 years in Zambia.

## Methods

This study constitutes a secondary analysis of microdata utilizing national-level data sourced from the Zambia Demographic and Health Survey (ZDHS) program. The ZDHS is a comprehensive, nationally representative household survey conducted by the Zambia Statistics Agency in collaboration with global partners, including ICF International and the United States Agency for International Development (USAID). The survey employs a two-stage sampling process, initially selecting enumeration areas (EAs) and subsequently households. The nature of the DHS data enables the comparison of variables over time, facilitating the monitoring of changes in indicators across various geographical regions [15].

The study included all individual files of participants aged 20 and above from selected households who had consented to take part in the research. The study adopted a complete case analysis were missing values were removed automatically at every point of analysist. Detailed methods employed in the DHS are comprehensively documented elsewhere [7]. For this specific study, we extracted all pertinent variables from the adolescent girls aged 15 to 19 years data files (individual recode) 2018 ZDHS datasets. The data under examination pertains to the population of adolescent's aged 15–19 years. Data collection took place from 18 July 2018 to 24 January 2019 [14]. Data was accessed 1st of October and 20th of November 2023. The authors in this study did not have access to information that could identify individual participants during or after data collection (see Fig 1).

### Dependent and independent variables

The focus of this study is adolescent fertility, specifically defined as the total number of children born to females aged 15 to 19 at the time of the survey. The variable of interest was evaluated by redefining the ZDHS variable related to the "total number of children ever born among female adolescents aged 15 to 19 years" into three categories: 0, 1, and 2 or more. Drawing from the literature, we organized potential fertility-influencing factors into three main categories: socio-economic, demographic, and individual-level factors [16–19]. Utilizing the DHS reference materials and data collection forms, we identified individual-level independent factors. The predictor variables encompassed the age of the woman (categorized as 15–19, 20–24, 25–29, 30–34, 35–39, 40–44 or 45–49 years), current marital status (categorized as not married or married), residence (categorized as urban or rural), education (classified as no education, primary, secondary, tertiary), contraceptive use (categorized as not using or using), employment status (categorized as employed or unemployed) and household wealth index (categorized as poorest, poorer, middle, richer or richest). Household wealth index is calculated using easy-to-collect data on a household's ownership of selected assets, such as televisions and bicycles; materials used for housing construction; and types of water access and sanitation facilities [20].

Community-level variables in this study were derived by aggregating individual-level data into clusters and encompassed community poverty, community education, community knowledge of family planning (FP) methods, and place of residence. These community-level

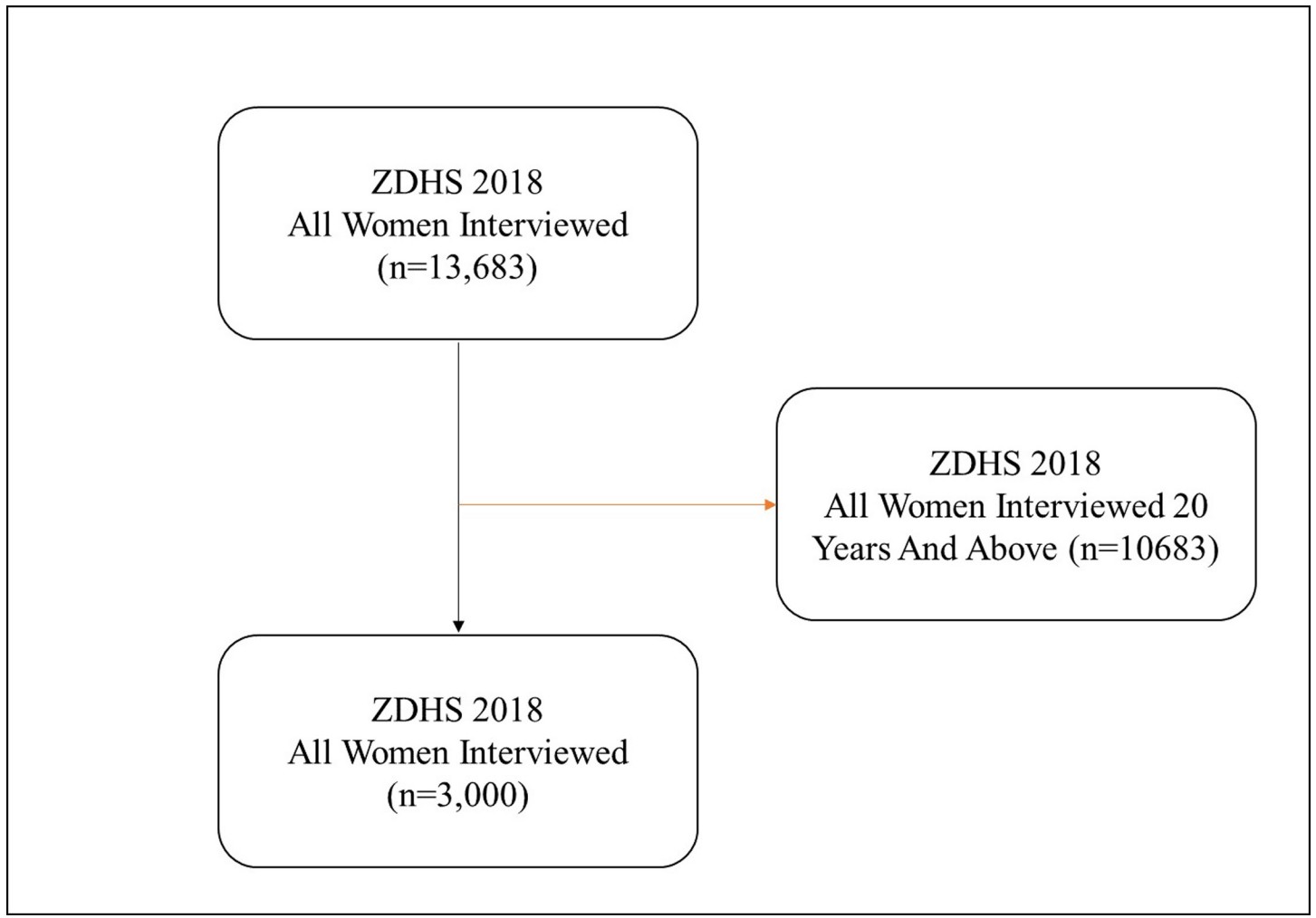

**Fig 1. Description of sample derivation criteria.** For this specific study, we extracted all pertinent variables from the women's data files (individual recode) 2018 ZDHS dataset. The data under examination pertains to the population of adolescent women in reproductive age group (15–19). Women 20 years and above were dropped in the study.

variables were dichotomized as either 'low' or 'high,' reflecting the extent of the phenomena under investigation at the cluster level. Place of residence and geographical region were maintained in their original categorizations. Place of residence played a pivotal role in the sample design, as it was utilized as a criterion to estimate the prevalence of key demographic and health indicators at the national level. It was categorized as either 'rural' or 'urban' and directly contributed to the description of community characteristics.

## Data analysis

For descriptive purposes, frequencies and percentages were computed for categorical variables using sample weighting for accurate representation. To determine association between the outcome variable ("ever given birth") and the categorical variables, the Uncorrelated Design Based Chi-square test (Rao–Scott Chi-square test) was used. This was selected to take into account clustering. The study furthermore utilized the survey multilevel mixed effect ordinal logistic regression (proportional odds model) to determine the factors associated with fertility

among adolescents. This approach accounted for the hierarchical structure of the data, with adolescents aged 15–19 nested within households and households nested within clusters. The outcome variable (ever given birth; no child, 1 child and 2+ children) is ordinal and using multinomial regression would give less efficient estimates. The study used an investigator led approach, all variables were selected from wide range literature. The probability F-test showed that the adopted model explained the outcome better than the null model (model without the explanatory variables), P<0.0001. In addition, the Brant test was utilized to examine whether the model had fulfilled the parallel lines assumption. A significant test statistic would suggest a violation of the parallel regression assumption. Consequently, the proportional odds model (p = 0.6083) was adopted with confidence, as it met the assumption. The log likelihood ratio test, Akaike Information Criteria (AIC) and the Bayesian Information Criteria (BIC) were sufficiently explored to select the best fit model. The variance for the predictors was less than 5 imposing no worry on multicolinearity in the model (see Table 1). Stata version 14.2 was used for the analysis.

The proportional odds model is expressed in the logit form as:

$$\ln\left(Y_j^i\right) = logit[\pi(x)] = \ln\left(\frac{\pi_j(x)}{1 - \pi_j(x)}\right) = \alpha_j + \left(-\beta_1 X_1 - \beta_2 X_2 - \cdots - \beta_p X_p\right)$$

Where, $\pi_j(x) = \pi(Y \leq j | x_1, x_2, \ldots, x_p)$, which is the probability of being at or below category j given a set of predictors, $j = 1, 2, \ldots, j-1$. $\alpha$ are the cut of points and $\beta_1, \beta_2, \ldots, \beta_p$ are logit coefficients. To estimate the ln(odds) of being at or below the *jth* category, the Proportional Odds

**Table 1. Multicolinearity test.**

| Variables | Variance Inflation Factor |
|---|---|
| Age of Adolescent | 1.54 |
| Education Level | 1.64 |
| Employment Status | 1.31 |
| Marital Status | 1.40 |
| Wealth Index | 3.25 |
| Contraceptive Use | 1.24 |
| Exposure to FP messages | 1.25 |
| Taught FP at H/F | 1.17 |
| Age at first sex | 1.54 |
| Frequency of listening to Radio | 1.14 |
| Frequency of listening to television | 1.83 |
| Residence | 1.89 |
| Community Poverty | 2.53 |
| Community Education | 1.85 |
| Community age at first birth | 1.03 |
| Community access to FP messages | 1.32 |
| Community employment status | 1.33 |
| Community ideal number of children | 1.10 |

model can be rewritten as:

$$logit[\pi(Y \leq j | x_1, x_2, \ldots, x_p)] = \ln\left(\frac{\pi(Y \leq j | x_1, x_2, \ldots, x_p)}{\pi(Y > j | x_1, x_2, \ldots, x_p)}\right)$$

$$= \alpha_j + (-\beta_1 X_1 - \beta_2 X_2 - \cdots - \beta_p X_p)$$

This model is based on the assumption of consistent effects represented by β for each logit [21, 22]. To find the magnitude and presence of relationships, odds ratios and their corresponding 95% confidence intervals were computed. The proportionality of odds for the dependent variable was evaluated using the Brant Test.

## Model selection

**Model 1** (competing model): Null model that is model without explanatory variable.

**Model 2** (Competing model): Individual level factors: Age of respondent, education, employment, contraceptive, exposure to FP, marital status, wealth index, taught FP at frequency of listening to a radio and frequency of listening to a television.

**Model 3** (accepted Model): Individual level factors: Age of respondent, education, employment, contraceptive, exposure to FP, marital status, wealth index, taught FP at frequency of listening to a radio and frequency of listening to a television, residential area, community poverty, community level of education, community age at first birth, community access to FP message, community employment and community ideal number of children.

## Ethical consideration

The 2018 Zambia Demographic Health Survey data survey protocols were approved by the ICF Institutional Review Board (IRB) under the ICF Project Number: 132989.0.000.ZM. DHS.02. Authorization to use the ZDHS data was obtained from ICF Macro, and the dataset, titled ZMIR71DTA, can be accessed at https://www.dhsprogram.com/data. The user diligently followed the provided instructions, emphasizing the confidential nature of the data and the importance of not attempting to identify any household or individual respondent interviewed in the survey (ensuring anonymity).

## Results

The study's findings reveal several notable patterns among the participants. Firstly, it is evident that a higher proportion of participants (95.61%) with no prior history of children were 15 years old. In contrast, a majority (37.26%) of participants who had two or more children were 19 years old. Furthermore, the data demonstrates that the majority of adolescents who gave birth between the ages of 15 to 17 comprised those who had only one or more children. Additionally, a significant proportion of these adolescents with one or at least children originated from rural areas (25.62% and 4.34% respectively). In terms of marital status, 60.2% of adolescents with a history of one childbirth were married, as opposed to those who were not. Similarly, a majority of adolescents with a history of multiple childbirths were married, accounting for 14.86%%. Moreover, the study's findings suggest a significant prevalence of adolescents from low-income households (poorest) with a high proportion of having 2 or more children (6.12%), in contrast to those adolescents from wealthier (richest) backgrounds (0.28%). This association was found to be statistically significant, with a p-value of less than 0.0001 as shown in Table 2 below.

**Table 2. Variables distribution and association of adolescent fertility in Zambia 2018.**

| Variables | Total number of children ever born (n = 3000) | | | |
|---|---|---|---|---|
| | No Child (n = 2277) | 1 Child (n = 634) | 2+ Children (n = 89) | P-value |
| **Demographic factors** | | | | |
| **Age** | | | | |
| 15 | 626 (95.61) | 26 (3.96) | 1 (0.09) | |
| 16 | 464 (87.45) | 64 (12.04) | 3 (0.51) | |
| 17 | 426 (77.21) | 117 (21.20) | 9 (1.60) | |
| 18 | 468 (64.85) | 225 (31.21) | 28 (3.94) | |
| 19 | 292 (53.75) | 202 (37.26) | 49 (8.99) | <0.0001[R***] |
| **Age of the adolescent at first birth** | | | | |
| Below 18 | Na | 158 (97.69) | 4 (2.31) | |
| 18 and above | Na | 476 (84.76) | 86 (15.24) | <0.0001[R***] |
| **Socio-economic factors** | | | | |
| **Educational level** | | | | |
| No education | 65 (66.48) | 25 (25.01) | 8 (8.51) | |
| Primary | 894 (69.69) | 325 (25.31) | 64 (5.00) | |
| Secondary | 1308 (81.34) | 283 (17.60) | 17 (1.04) | |
| Tertiary | 8 (83.27) | 2 (16.73) | 0(0.00) | <0.0001[R***] |
| **Current Marital Status** | | | | |
| Not married | 2169 (84.44) | 374 (14.58) | 25 (0.98) | |
| Married | 108 (24.93) | 260 (60.20) | 64 (14.86) | <0.0001[R***] |
| **Currently working** | | | | |
| No | 1971 (79.57) | 452 (18.26) | 54 (2.17) | |
| Yes | 305 (58.39) | 182 (34.78) | 36 (6.82) | <0.0001[R***] |
| **At health facility told of FP** | | | | |
| No | 728 (72.70) | 242 (24.17) | 31 (3.13) | |
| Yes | 121 (32.04) | 225 (59.41) | 32 (8.55) | <0.0001[R***] |
| **Wealth index** | | | | |
| Poorest | 321 (62.92) | 158 (30.97) | 31 (6.12) | |
| Poorer | 363 (67.18) | 148 (27.34) | 30 (5.49) | |
| Middle | 422 (72.20) | 149 (25.55) | 13 (2.25) | |
| Richer | 502 (76.65) | 140 (21.31) | 13 (2.03) | |
| richest | 668 (94.17) | 39 (5.56) | 1.96 (0.28) | <0.0001[R***] |
| **Behavioral factors** | | | | |
| **Current contraceptive use** | | | | |
| Not using | 2211 (83.84) | 380 (14.42) | 46 (1.74) | |
| Using | 66 (18.12) | 2544 (69.91) | 44 (11.97) | <0.0001[R***] |
| **Age at first sex** | | | | |
| **Not had** | 1510 (100.00) | 0 (0) | 0 (0) | |
| 9–12 | 29 (54.35) | 19 (36.35) | 5 (9.30) | |
| 13–16 | 479 (45.56) | 492 (46.80) | 80 (7.64) | |
| 17–19 | 257 (67.09) | 123 (31.87) | 4 (1.04) | <0.0001[R***] |
| **Frequency of listening to radio** | | | | |
| Not at all | 1231 (72.10) | 412 (24.11) | 65 (3.79) | |
| <once a week | 336 (82.11) | 68 (16.53) | 6 (1.35) | |
| ≥ once a week | 409 (81.96) | 79 (15.86) | 11 (2.18) | |
| Almost everyday | 302 (78.15) | 76 (19.73) | 8 (2.12) | 0.0004[R**] |
| **Frequency of watching to television** | | | | |

(*Continued*)

**Table 2.** (Continued)

| Variables | No Child (n = 2277) | 1 Child (n = 634) | 2+ Children (n = 89) | P-value |
|---|---|---|---|---|
| **Total number of children ever born (n = 3000)** | | | | |
| **Demographic factors** | | | | |
| Not at all | 1164 (68.22) | 466 (27.29) | 77 (4.48) | |
| <once a week | 147 (77.39) | 37 (19.67) | 6 (2.94) | |
| ≥ once a week | 250 (84.42) | 41 (13.92) | 5 (1.66) | |
| Almost everyday | 716 (88.56) | 90 (11.15) | 2 (0.29) | <0.0001[R***] |
| **Community level factors** | | | | |
| **Province** | | | | |
| Central | 218 (73.59) | 66 (22.22) | 12 (4.19) | |
| Copperbelt | 398 (81.19) | 83 (16.99) | 9 (1.82) | |
| Eastern | 231 (67.61) | 95 (27.72) | 16 (4.68) | |
| Luapula | 195 (77.02) | 48 (8.76) | 11 (4.22) | |
| Lusaka | 420 (88.47) | 50 (10.55) | 5 (0.98) | |
| Muchinga | 148 (77.68) | 34 (17.95) | 8 (4.37) | |
| Northern | 195 (78.35) | 51 (20.41) | 3 (1.23) | |
| North western | 130 (69.52) | 47 (25.01) | 10 (5.46) | |
| Southern | 214 (65.32) | 108 (32.90) | 6 (1.78) | |
| Western | 127 (66.95) | 53 (28.18) | 9 (4.87) | <0.0001[R***] |
| **Residence** | | | | |
| Urban | 1102 (83.27) | 205 (15.47) | 17 (1.25) | |
| Rural | 1175 (70.04) | 430 (25.62) | 73 (4.34) | <0.0001[R***] |
| **Community poverty** | | | | |
| Low | 1420 (81.16) | 299 (17.10) | 30 (1.74) | |
| Medium | 269 (71.96) | 92 (24.44) | 14 (3.60) | |
| High | 587 (67.01) | 244 (27.81) | 45 (5.18) | <0.0001[R***] |
| **Community education** | | | | |
| Low | 1608 (79.08) | 386 (18.96) | 40 (1.96) | |
| Medium | 252 (70.82) | 97 (27.23) | 7 (1.95) | |
| High | 416 (68.16) | 152 (24.86) | 43 (6.98) | <0.0001[R***] |
| **Community age birth** | | | | |
| Low | 268 (76.78) | 75 (21.50) | 6 (1.72) | |
| Medium | 156 (64.19) | 86 (35.28) | 1 (0.53) | |
| High | 1071 (65.84) | 474 (29.11) | 82 (5.04) | <0.0001[R***] |
| **Community employment** | | | | |
| Low | 136 (67.34) | 56 (27.55) | 10 (5.07) | |
| Medium | 129 (56.77) | 85 (37.53) | 13 (5.67) | <0.0001[R***] |
| High | 2011 (78.24) | 493 (19.19) | 66 (2.58) | |
| **Community access to FP messages** | | | | |
| Low | 80 (79.67) | 18 (18.10) | 2 (2.22) | |
| High | 119 (85.56) | 18 (12.74) | 2 (1.70) | |
| Medium | 2078 (75.25) | 598 (21.68) | 85 (3.07) | 0.2929 [R] |
| **Community ideal number of children** | | | | |
| Low | 1841 (77.15) | 490 (20.54) | 55 (2.32) | |
| Medium | 148 (74.72) | 41 (20.64) | 9 (4.64) | |
| High | 287 (69.10) | 103 (24.91) | 25 (5.99) | 0.0007 [R***] |

R = Rao–Scott Chi-square test

*** = P < 0.05

### Adolescent girls with no history of a childbirth

In urban settings, the prevalence of adolescents without a history of childbirth was slightly higher than in rural settings, with the exception of the southern province. In the southern province, a notable proportion of adolescent girls in rural areas reported no history of childbirth compared to their urban counterparts. Additionally, a significant difference in the population of adolescents without a history of childbirth was observed in the eastern province, as depicted in Fig 2.

### Adolescent girls with a history of one childbirth

Additionally, the study's findings indicate a higher count of adolescents with a history of one childbirth hailing from rural settings, with the exception of the southern province, where there is a greater proportion of adolescents in urban settings who have experienced childbirth. Furthermore, Luapula province documented the lowest percentage of adolescents with a history of one childbirth, and in urban areas, the lowest proportion was observed in Lusaka province as shown in Fig 3.

### Adolescent girls with history of at least two childbirths

The study's results in Fig 4 reveal a higher percentage of adolescents in rural settings with a history of at least two childbirths across all provinces, except for the Northern Province. Noteworthy variations were observed in the central, eastern, Luapula, Lusaka, and northwestern provinces. Specifically, the Northern Province exhibited a higher proportion of adolescents in rural settings with a record of at least two childbirths, followed by the central province. In urban settings, the Western Province took the lead, followed by Muchinga.

### Multilevel mixed effect ordinal logistic regression

The study investigated the factors associated with adolescent fertility using the multilevel mixed effect ordinal logistic regression. The findings suggest that a year increase in the age of

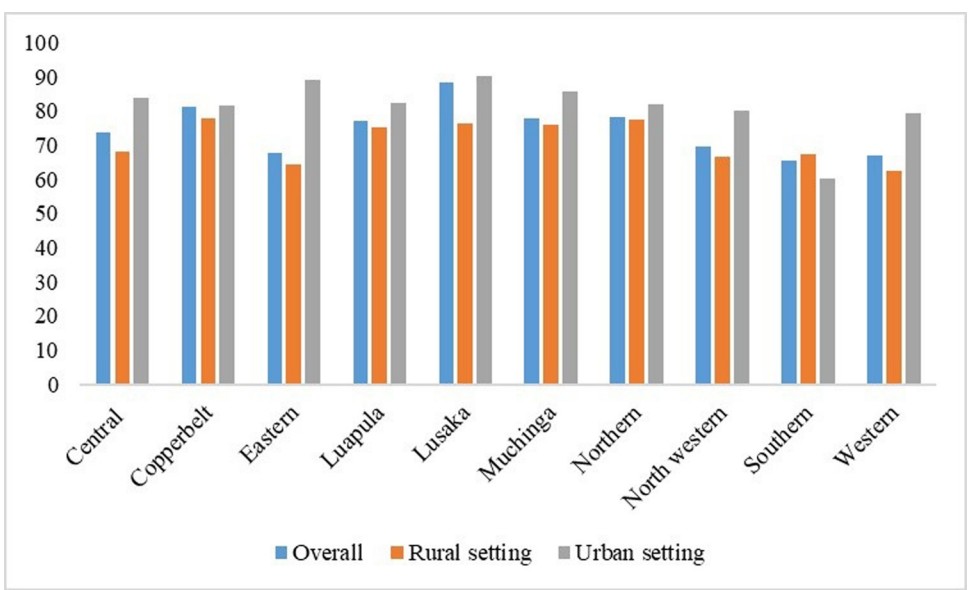

**Fig 2. Adolescent girls with no history of a childbirth.** Generated by excel.

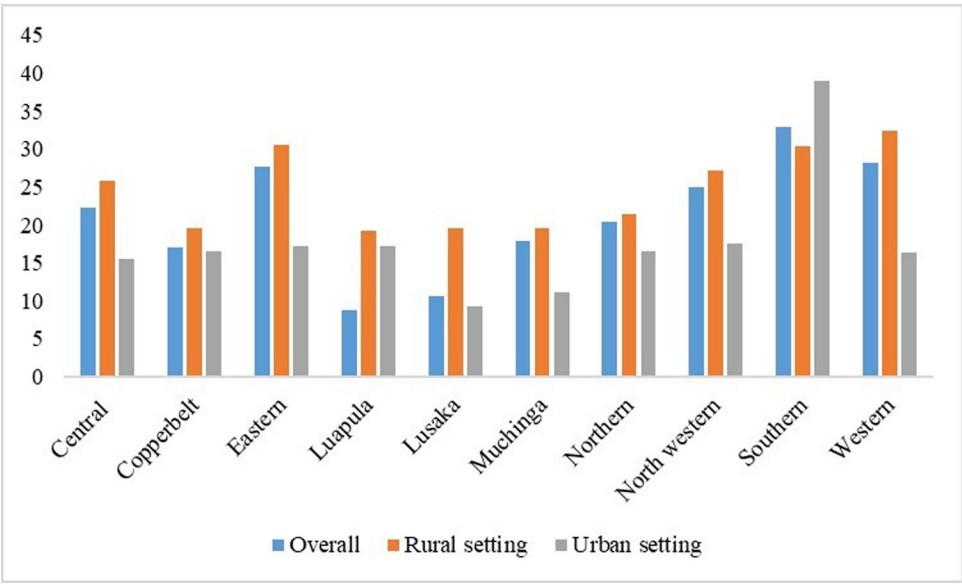

**Fig 3. Adolescent girls with a history of one childbirth.** Generated by excel.

adolescents increased the odds of being beyond a particular category of fertility (Multiple child births), given the effects of all other predictors are held constant. In other words, older adolescents were associated with increased odds of having multiple children, (AOR, 1.50; 95% CI, 1.31–1.72; P<0.0001) and this was statistically significant. In the same vein, adolescents with a higher level of education (primary, secondary and tertiary education) had reduced odds of being beyond a particular fertility category compared to those with no level of education (AOR, 0.47; 95% CI, 0.23–0.97; AOR, 0.2; 95% CI, 0.10–0.47; AOR, 0.03; 95% CI, 0.00–0.54 respectively).

Furthermore adolescents who were married, used contraceptives, coming from the poorer backgrounds or had an exposure to family planning (FP) messages had increased odds of having multiple children (being beyond a particular fertility category) (AOR, 2.56; 95% CI, 1.78–3.67; AOR, 3.09; 95% CI, 2.20–4.32; AOR, 1.68; 95% CI, 1.11–2.53 and AOR, 1.18; 95% CI, 1.13–1.22 respectively). Similarly adolescent told about FP at a health facility had increased odds of 2.77 of being beyond a particular category of fertility (higher number of births) compared to those were not told (95% CI, 2.04–3.76). Holding all things constant, a year increase in age at first sex increased the odds of having multiple children by a factor of 1.18 times (AOR, 1.18; 95% CI, 1.13–1.22). Among the contextual factors, that is residential area, community poverty, community level of education, community age at first birth, community access to FP message, community employment and community idea number of children), only community age at first birth was predicting multiple births among adolescents. In other words, adolescents from a high community age at first birth had increased odds of being beyond a particular category of fertility compared to those from a low community age at first birth (AOR, 1.59; 95% CI, 1.01–2.52), see Table 3.

## Discussion

This study analyzed the predictors of fertility among older adolescents aged between 15 and 19 in Zambia. Utilizing a multilevel ordinal logistic regression on data from the 2018 Zambia Demographic and Health Surveys, the study aimed to gain deeper insights into the factors

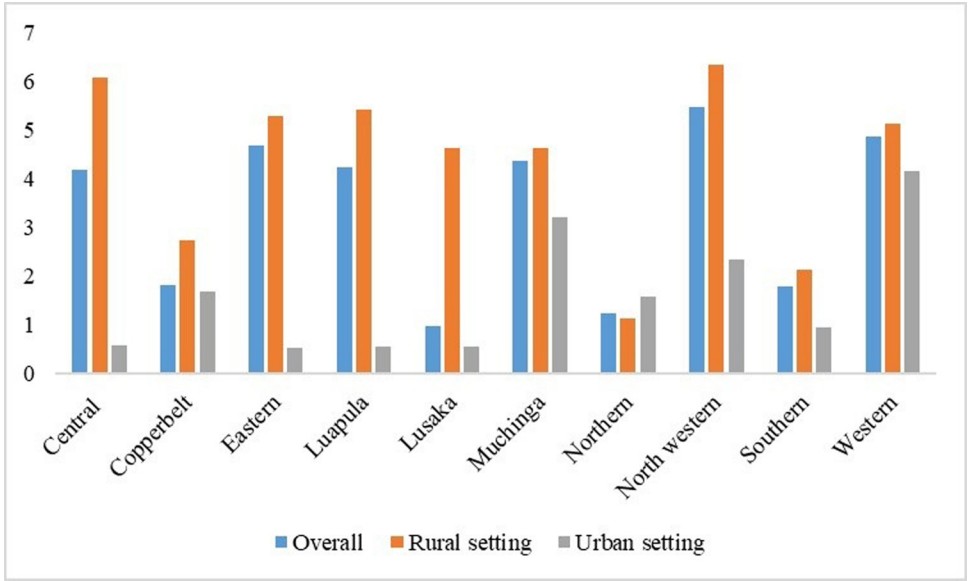

**Fig 4. Adolescent girls with history of at least two childbirths.** Generated by excel.

contributing to elevated adolescent fertility in this age group. The study identified demographic and socio-economic disparities in adolescent fertility, revealing that a majority of adolescents with one or more children were aged 19. About 60% of married adolescents had one child, and approximately 14% in rural areas had more than one child. Rural areas showed significant increases in adolescents with multiple children. Furthermore, those with no formal education, married, residing in rural areas, low-income, or unemployed were more likely to have multiple children. This aligns with studies in Kenya, Nigeria, Malawi, and Tanzania [23–26]. However, the study noted that the percentage of adolescents with one child was higher than in Malawi, Uganda, Tanzania, Ethiopia, Rwanda, Eritrea, and Ghana [27–30].

The study revealed that certain individual factors such as education, marital status, wealth index, contraceptive use, exposure to family planning messages, and receiving family planning education at health facilities were significantly associated with higher adolescent fertility in Zambia. Notably, only community age at first birth emerged as a significant predictor of higher adolescent fertility. Among the models considered, model three, which incorporated both individual and community-level variables, was accepted. This decision was based on its superior fit, as evidenced by smaller values of AIC, BIC, and Log likelihood in comparison to the null model (without individual and community level variables) and model two (including only individual-level variables).

Furthermore, the study findings indicate that older adolescents face an increased risk of having more than one child. This aligns with observations from studies conducted in Kenya, Nigeria and Ghana, where it was noted that older adolescents are more susceptible to higher fertility compared to their younger counterparts [23, 27, 31]. A similar trend was identified in a study conducted in Indonesia, revealing a positive correlation between the age of adolescents and birthrate [32]. Similarly, our study revealed that adolescents who were older at the time of their first sexual encounter and those from communities with a higher community age at birth were more likely to have more than one child. This finding contradicts a study conducted in Uganda, which found that individuals who engaged in early sexual activity and delayed

**Table 3. Multilevel mixed effect ordinal logistic regression.**

| Variables | Model 1 | Model 2 | Model 3 |
|---|---|---|---|
| | OR | AOR | AOR |
| **Intercept** | | | |
| Intercept 1 | 3.42 (3.07–3.81)*** | 8.13 (5.81–10.44)*** | 8.77 (6.07–11.47)*** |
| Intercept 2 | 3.58 (3.36–3.81)*** | 12.03 (9.60–14.46)*** | 12.78 (10.01–15.56)*** |
| **Socioeconomic and demographic Factors** | | | |
| **Age of Adolescent** | | 1.39 (1.22–1.60)*** | 1.50 (1.31–1.72)*** |
| **Education Level** | | | |
| No Education | | Ref (1) | Ref (1) |
| Primary | | 0.50 (0.24–1.03) | 0.47 (0.23–0.97)* |
| Secondary | | 0.26 (0.12–0.55)*** | 0.21 (0.10–0.47)*** |
| Tertiary | | 0.06 (0.06–0.90)* | 0.03 (0.00–0.54)** |
| **Employment Status** | | | |
| Not employed | | Ref (1) | Ref (1) |
| Employed | | 1.15 (0.83–1.60) | 1.08 (0.75–1.55) |
| **Marital Status** | | | |
| Not married | | Ref (1) | Ref (1) |
| Married | | 2.49 (1.76–3.52)*** | 2.56 (1.78–3.67)*** |
| **Wealth Index** | | | |
| Poorest | | Ref (1) | Ref (1) |
| Poorer | | 1.69 (1.13–2.53)* | 1.68 (1.11–2.53)** |
| Middle | | 0.99 (0.65–1.52) | 0.86 (0.51–1.44) |
| Richer | | 0.99 (0.58–1.70) | 0.85 (0.44–1.67) |
| Richest | | 0.68 (0.33–1.42) | 0.74 (0.30–1.81) |
| **Behavioral factors** | | | |
| **Contraceptive Use** | | | |
| No | | Ref (1) | Ref (1) |
| Yes | | 3.99 (2.84–5.62)*** | 3.09 (2.20–4.32)*** |
| **Exposure to FP messages** | | | |
| No | | Ref (1) | Ref (1) |
| Yes | | 0.62 (0.17–2.29) | 1.18 (1.13–1.22)*** |
| **Taught FP at H/F** | | | |
| No | | Ref (1) | Ref (1) |
| Yes | | 2.86 (2.09–3.92)*** | 2.77 (2.04–3.76)*** |
| **Age at first sex** | | 1.18 (1.13–1.22)*** | 1.18 (1.13–1.22)*** |
| **Frequency of listening to Radio** | | | |
| Not at all | | Ref (1) | Ref (1) |
| Less than once a week | | 0.62 (0.39–0.98)* | 0.64 (0.40–1.02) |
| At least once a week | | 0.87 (0.56–1.36) | 0.94 (0.60–1.49) |
| Almost everyday | | 0.99 (0.63–1.54) | 0.91 (0.57–1.44) |
| **Frequency of listening to television** | | | |
| Not at all | | Ref (1) | Ref (1) |
| Less than once a week | | 0.73 (0.38–1.40) | 0.74 (0.38–1.43) |
| At least once a week | | 0.46 (0.24–0.87) | 0.56 (0.29–1.08) |
| Almost everyday | | 0.71 (0.40–1.26) | 0.85 (0.47–1.55) |
| **Contextual Factors** | | | |
| **Residential area** | | | |
| Urban | | | Ref (1) |
| Rural | | | 1.02 (0.77–1.87) |

(*Continued*)

**Table 3.** (Continued)

| Variables | Model 1 | Model 2 | Model 3 |
|---|---|---|---|
|  | OR | AOR | AOR |
| **Community Poverty** |  |  |  |
| Low |  |  | Ref (1) |
| Medium |  |  | 0.89 (0.52–1.50) |
| High |  |  | 0.94 (0.55–1.60) |
| **Community Education** |  |  |  |
| Low |  |  | Ref (1) |
| Medium |  |  | 0.67 (0.41–1.07) |
| High |  |  | 0.73 (0.47–1.15) |
| **Community age at first birth** |  |  |  |
| Low |  |  | Ref (1) |
| Medium |  |  | 1.24 (0.67–2.28) |
| High |  |  | 1.59 (1.01–2.52)* |
| **Community access to FP messages** |  |  |  |
| Low |  |  | Ref (1) |
| High |  |  | 0.43 (0.11–1.66) |
| Medium |  |  | 0.55 (0.23–1.33) |
|  |  |  |  |
| **Community employment status** |  |  |  |
| Low |  |  | Ref (1) |
| Medium |  |  | 1.91 (0.99–3.97) |
| High |  |  | 1.22 (0.71–2.09) |
| **Community ideal number of children** |  |  |  |
| Low |  |  | Ref (1) |
| Medium |  |  | 0.77 (0.46–1.28) |
| High |  |  | 0.94 (0.63–1.39) |
| **Random effects** |  |  |  |
| Variance (SE) | 0.1 | 0.19 | - |
| VPC (%) | 0.4 | 0.31 | 0 |
| AIC | 3985.62 | 1534.32 | 1379.77 |
| BIC | 4003.75 | 1656.18 | 1557.63 |
| Log likelihood | -1989.81 | -744.16 | -654.88 |
| VIF | Na | <5 | <5 |

*** = P-value ≤ 0.01

** = 0.01<P-value ≤ 0.03

* = 0.03 <P-value <0.05

childbirth were less likely to experience higher fertility compared to those who underwent these events early [23, 33].

Our findings indicate that adolescent girls who completed primary, secondary, and tertiary education were less likely to have more children compared to their counterparts with no formal education. This observation is consistent with studies conducted in Nigeria, where it was established that adolescents aged 15–19 with primary education or higher exhibited lower risks of having children in comparison to adolescents with no formal education [25, 34]. This suggests that education plays a crucial role in enhancing autonomy, decision-making abilities, and economic independence. Consequently, it contributes to delaying marriage and sexual

debut while increasing knowledge about contraception, as noted in studies such as those by Mohr et al. [35].

Marital status in the study similarly predicted adolescent fertility. The findings suggests that married adolescent girls had a higher chance of having multiple children compared to those who were not. This aligns with research conducted in various African countries, including Burundi, Ethiopia, Nigeria, Kenya, Congo, and Central Africa, where early marriages have been identified as a significant factor contributing to high fertility rates [23, 36–38]. This implies that early child marriages play a central role in driving increased adolescent fertility. The pressure exerted on married adolescents by their families and partners to start their own families contributes to this phenomenon. Teenage pregnancies resulting from early marriages pose heightened risks, including increased chances of maternal mortality during childbirth and elevated rates of neonatal and infant deaths among young mothers. Girls aged between 15 and 19 face double the likelihood of dying during childbirth compared to women aged 20 and above. This risk is further compounded by factors such as HIV, making complications during pregnancy and childbirth the primary cause of death for young women in this age group. Additionally, pregnant adolescents are more susceptible to serious health issues such as eclampsia, puerperal endometritis, and systemic infections compared to adults [39].

Furthermore adolescents who used contraceptives, were exposed to family planning (FP) or received FP education at health facilities were more likely to have more than one child compared to those who did not use contraceptives. This aligns with research conducted in Kenya and Zambia, which indicated that adolescents using contraceptives were at a higher risk of experiencing higher fertility [23, 40]. This phenomenon could be attributed to the observation that adolescents often initiate contraceptive use after having their first child. Furthermore, a contributing factor may be the cultural norms that discourage having children out of wedlock, particularly prevalent in rural areas, leading many girls who become pregnant to enter early marriages [41].

In contrasts to the use of contraceptives, exposure to family planning (FP) or receiving FP the findings in Ethiopia, Namibia, and Burundi, where research indicated that teenagers who receive sexual education through media channels, such as television and radio, are likely to abstain from sexual intercourse due to knowledge about the dangers of early engagement, such as sexually transmitted infections and risky pregnancies leading to potential harm [36, 42, 43]. This inconsistency might be attributed to the perceptions held by young people that family planning messages are primarily intended for older married individuals and do not apply to them, leading them to disregard or avoid such messages [37].

Residence did not emerge as a significant predictor of adolescent fertility in the study, a result that aligns unexpectedly with studies conducted in Ethiopia, Uganda, and Mexico. These studies similarly found that the place of residence was not a significant predictor of higher adolescent fertility in Zambia [44, 45]. In contrasts to the findings in the descriptive results that shows a higher proportion of adolescents with at least one childbirth residing in rural areas across majority of the provinces. The differences are much wider among adolescents with at least two multiple childbirths.

Furthermore, as a way combating fertility among adolescents the government of Zambia has introduced Comprehensive Sexuality Education (CSE) in schools. Additionally, the Government of Sweden, through the Centre for People (PeaCe) health program, has supported the Ministry of Health (MoH) [46]. A key strategy of the program aims to reduce adolescent pregnancy by increasing access to quality Adolescent Health (ADH) information and promoting behavior change through outreach and facility-based engagement models. With support from the Clinton Health Access Initiative (CHAI), MoH has trained healthcare workers and peer educators (PEs) in selected provinces to provide people-centered adolescent health services,

including sexual and reproductive health (SRH) and information services, both at the community and facility levels [47].

## Study strength and study limitations

This study draws strength from the utilization of national data, providing a representative sample of the adolescent female population aged 15 to 19 in Zambia. Consequently, the study's findings are applicable and can be generalized to the specified target population of adolescent girls within this age range. However, it is essential to acknowledge the study's limitations. The reliance on the latest Zambia Demographic and Health Survey (ZDHS) dataset from 2018 follows a cross-sectional study design, implying that the results indicate correlation rather than causation between the outcome of interest and individual or contextual factors.

Additionally, caution is advised when extending the findings to the broader adolescent age group of 10 to 19 years. Moreover, the contextual factors utilized in the study are derived from the ZDHS, potentially limiting their ability to fully capture the community experience. It is important to acknowledge the potential presence of recall bias in the study. There is a possibility of mothers misreporting essential information, such as the total number of children ever born. To mitigate the recall bias, performed a thorough data cleaning to identify and address inconsistencies or outliers. It is essential to recognize these factors for a nuanced interpretation of the study's findings.

## Conclusion and policy implications

The study has shown disparities in adolescent fertility across sociodemographic groups, and further emphasizing the importance of understanding contributing factors for effective sexual reproductive health policies. Education emerged as a protective factor, with completion of primary, secondary, and tertiary education associated with lower likelihoods of having more than one child. Marital status played a significant role, with married adolescent girls having a higher likelihood of multiple children, highlighting the impact of early marriages on fertility rates. Residence did not predict adolescent fertility. Furthermore, the study provided valuable insights into the multifaceted nature of adolescent fertility in Zambia, emphasizing education, marital status, and considerations of contraceptive use and cultural influences.

To address the challenge of adolescent fertility in Zambia, a comprehensive policy approach is essential. The government's initiatives, such as the introduction of Comprehensive Sexuality Education (CSE) in schools, and support from international partners like the Government of Sweden's Centre for People (PeaCe) health program and the Clinton Health Access Initiative (CHAI), demonstrate commitment to tackling this issue. Policies should prioritize expanding access to Adolescent Health (ADH) information through innovative outreach and facility-based engagement models. Education-focused initiatives, including awareness campaigns, should promote primary, secondary, and tertiary education for adolescents. Strategies to prevent early marriages, including legal reforms and community awareness, are crucial. Reproductive health education must be enhanced, providing accurate information on contraception, family planning, and the consequences of early childbearing. Community-based interventions should be tailored to address contextual factors, engaging local leaders and considering cultural nuances. Continuous research and monitoring are vital to inform evidence-based policies, ensuring effectiveness. This multifaceted approach aims to reduce adolescent fertility comprehensively, with a particular emphasis on rural areas where the need may be more pronounced.

## Acknowledgments

We wish to express our sincere gratitude to the Zambia Statistics Agency and the DHS Program for granting permission to utilize the 2018 ZDHS dataset.

## Author Contributions

**Conceptualization:** Samson Shumba.

**Data curation:** Samson Shumba.

**Formal analysis:** Samson Shumba.

**Methodology:** Samson Shumba, Vanessa Moonga, Thomas Osman Miyoba, Stephen Jere, Jessy Mutale Nkonde, Peter Mumba.

**Software:** Samson Shumba.

**Visualization:** Peter Mumba.

**Writing – original draft:** Samson Shumba, Vanessa Moonga, Thomas Osman Miyoba, Stephen Jere, Jessy Mutale Nkonde, Peter Mumba.

**Writing – review & editing:** Samson Shumba, Vanessa Moonga, Thomas Osman Miyoba, Stephen Jere, Jessy Mutale Nkonde, Peter Mumba.

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
