## [Decision Letter · Decision Letter 0]

14 Jan 2024

PGPH-D-23-02336

Socio-economic disparities and predictors of fertility among adolescents aged 15 to 19 in Zambia: Evidence from the Zambia demographic and health survey (2018)

Dear Dr. Samson Shumba,

Thank you for submitting your manuscript to PLOS Global Public Health. After careful consideration, we feel that it has merit but does not fully meet PLOS Global Public Health’s publication criteria as it currently stands. Therefore, we invite you to submit a revised version of the manuscript that addresses the points raised during the review process.

Please address the following comments and those of reviewers.

a) Specify the independent and dependent variables in the abstract.

b) Mention few more key words.

c) The rationale of this study requires more development. Please review relevant literature, leading to your paper aim.

d) How did you select the independent variables? Based on review of the literature? If yes, please state it with citations.

e) Ethics section can be shorter as you applied the secondary sources of data, which do not require ethics approval.

f) The first paragraph of the discussion section requires revision. Please state and summarise your findings here.

g) Delete justification as it is in a wrong place; it should be in the introduction. Rather state the limitations and strengths of your study here.

h) Conclusions and policy implications?

We look forward to receiving your revised manuscript.

Kind regards,

Md Nazmul Huda, PhD

Academic Editor

Journal Requirements:

Additional Editor Comments (if provided):

Reviewers' comments:

Reviewer's Responses to Questions

**Comments to the Author**

1. Does this manuscript meet PLOS Global Public Health’s publication criteria? Is the manuscript technically sound, and do the data support the conclusions? The manuscript must describe methodologically and ethically rigorous research with conclusions that are appropriately drawn based on the data presented.

Reviewer #1: Partly

Reviewer #2: Yes

2. Has the statistical analysis been performed appropriately and rigorously?

Reviewer #1: Yes

Reviewer #2: Yes

3. Have the authors made all data underlying the findings in their manuscript fully available (please refer to the Data Availability Statement at the start of the manuscript PDF file)?

Reviewer #1: Yes

Reviewer #2: Yes

4. Is the manuscript presented in an intelligible fashion and written in standard English?

Reviewer #1: Yes

Reviewer #2: Yes

5. Review Comments to the Author

Reviewer #1: The article “Socio-economic disparities and predictors of fertility among adolescents aged 15 to 19 in Zambia: Evidence from the Zambia demographic and health survey (2018)” was designed to investigate the socio-economic disparities and associated factors of fertility among adolescents aged 15 to 19 years in Zambia. Overall, the article provides some unique insights. There are however major issues that must be addressed within this manuscript to fully address the topic being investigated. Major conceptual and analytical issues were noticed in the Methods, Results, and Discussion sections.

Abstract: Missing aim or hypothesis to be tested and details in design.

Introduction:

More detailed statistics are needed after the statement, “Zambia, has grappled with elevated fertility rates…”

Serves more as a review of ideas surrounding childbirth and potential individual and community level factors than as a means to allow for deductive reasoning that would allow for a hypothesis to be developed that is necessary for conducting the analysis that you have indicated performing.

Ignoring numerous studies have already shown the interaction of childbirth, individual and community level factors.

Missing a clear definition of the main variable of interest “fertility,” and how the authors utilize this term in the study.

Methods

Manuscript lacks consistency in the use of key terms, specifically for the population of interest e.g. adolescent’s aged 15–19 versus women.

A brief description is needed for the Bongaarts proximate determinants of fertility.

Missing sample size details (3,112 only is mentioned in the abstract). The Zambia Demographic and Health Survey seems to have more than 3,112.

Missing details necessary to explain selection process of data included and excluded or how you obtained information.

Missing information if the sample weighting option was used/not used for such national representative data? And why?

How were you able to control the existing bias in responses?

Missing explanation of inclusion and exclusion of responses e.g. missing values

Dependent and independent variables:

The description and use of the main dependent variable depicts a faulty message, and it may not reflect the actual meaning of the variable, “ever given birth/children ever born”. The variable appears to be more of "Yes, No" question instead of multilevel responses that are used in the manuscript, “No Child, 1 Child, and 2+ Children”, which indicates to the number of children.

How did you handle measures to account for biased responses?

Missing details on included measures and how they were calculated e.g. wealth index.

Data Analysis:

Incomplete description of methods

Missing how you established parameters for Chi-square

Reporting on odds ratio, but no indication for calculating an odds ratio or how you went about adjust the odds ratio to indicate why you needed to adjust that odds ratio.

Results:

Cannot critically analyze report based on faulty/unclear utilization of variables, specifically the variable of interest “fertility”.

Tables are clear and easy to follow.

Model 1 was labeled as “AOR”, however, it is unclear whether this model was adjusted for other variables.

Have you considered using the age specific fertility rate (ASFR) or average/mean number of children ever born (MCEB) instead of just number of children?

Discussion:

Cannot critically discuss due to faulty/unclear utilization of variables and subsequently analysis and results being offered.

Missing the inductive reasoning that would provide a means to draw a conclusion and offer insight into how findings fit into/contribute to the larger base of knowledge that we already have.

More results need to be emphasized, discussed and compared e.g. children ever born vs wealth index.

Biased responses were not indicated in the limitation.

Missing detailed information of current reproductive health policies, and recommended policies to be implemented.

Reviewer #2: The article provides substantial information regarding the topic. There are a couple of sentences which need to be referenced. For instance, 'Studies have shown that adolescent mothers face high risks of eclampsia...

6. PLOS authors have the option to publish the peer review history of their article (what does this mean?). If published, this will include your full peer review and any attached files.

**Do you want your identity to be public for this peer review?** For information about this choice, including consent withdrawal, please see our Privacy Policy.

Reviewer #1: No

Reviewer #2: **Yes: **Shazia Khalid

---

## [Editor Report · Decision Letter 1]

28 Feb 2024

Socio-economic disparities and predictors of fertility among adolescents aged 15 to 19 in Zambia: Evidence from the Zambia demographic and health survey (2018)

PGPH-D-23-02336R1

Dear Dr Samson Shumba,

We are pleased to inform you that your manuscript 'Socio-economic disparities and predictors of fertility among adolescents aged 15 to 19 in Zambia: Evidence from the Zambia demographic and health survey (2018)' has been provisionally accepted for publication in PLOS Global Public Health.

Best regards,

Md Nazmul Huda, BSS, MSS, PhD

Academic Editor